# Integrating UAV Technology in an Ecological Monitoring System for Community Wildlife Management Areas in Tanzania

**Lazaro J. Mangewa [1,2,*], Patrick A. Ndakidemi [1] and Linus K. Munishi [1]**

1   School of Life Sciences and Bio-Engineering (LISBE), Nelson Mandela African Institution of Science and Technology, Arusha P.O. Box 447, Tanzania; patrick.ndakidemi@nm-aist.ac.tz (P.A.N.); linus.munishi@nm-aist.ac.tz (L.K.M.)
2   Centre for Research, Agricultural Advancement, Teaching Excellence and Sustainability in Food and Nutrition Security (CREATES-FNS), Nelson Mandela African Institution of Science and Technology, Arusha P.O. Box 447, Tanzania
*   Correspondence: mangewal@nm-aist.ac.tz

**Abstract:** Unmanned aerial vehicles (UAV) have recently emerged as a new remote sensing aerial platform, and they are seemingly advancing real-time data generation. Nonetheless, considerable uncertainties remain in the extent to which wildlife managers can integrate UAVs into ecological monitoring systems for wildlife and their habitats. In this review, we discuss the recent progress and gaps in UAV use in wildlife conservation and management. The review notes that there is scanty information on UAV use in ecological monitoring of medium-to-large mammals found in groups in heterogeneous habitats. We also explore the need and extent to which the technology can be integrated into ecological monitoring systems for mammals in heterogeneous habitats and in topographically-challenging community wildlife-management areas, as a complementary platform to the traditional techniques. Based on its ability to provide high-resolution images in real-time, further experiments on its wider use in the ecological monitoring of wildlife on a spatiotemporal scale are important. The experimentation outputs will make the UAV a very reliable remote sensing platform that addresses the challenges facing conventional techniques.

**Keywords:** UAV; remote sensing; conventional ecological monitoring techniques; satellite platforms; medium-to-large mammals; UAV-based habitat assessment

## 1. Introduction

Real-time ecological data is critical for effective and efficient monitoring of wildlife populations and habitats to deal with the threats that face wildlife [1–5]. Recently, unmanned aerial vehicles (UAVs), also known as drones or rapidly piloted aircrafts (RPAs), have emerged as an alternative aerial survey platform for fauna and their habitats, and they advance the generation of real-time data [5,6]. Notwithstanding the benefits of this new technology, considerable questions remain about the extent of its integration into ecological monitoring systems for community wildlife management areas (WMAs) on a spatiotemporal scale. The system is important for landscapes with heterogeneous habitats and diverse topographical features. Habitat heterogeneity is a common characteristic of most wildlife management areas in the tropics [7,8]. Such areas comprise bare lands, grasslands, shrublands, open or closed woodlands, and forests. The density of each habitat or a mixture of these habitats has implications for the detection probability. The UAV is easily used to capture images in open habitats [9]. However, there are inadequate studies in mixed habitats on a spatiotemporal scale. It is, thus, important to ascertain the use of UAVs for different and mixed habitat types across seasons.

The collection of adequate ecological information about the mammals and their habitats is crucial for effective monitoring and overall management practices. An appropriate application of improved technologies, such as UAVs, to produce adequate and relevant ecological data for effective conservation and management of wildlife is a central goal of conservation [10]. The use of data from UAV images to guide conservation and management actions is expanding. However, researchers need to understand the challenges associated with the technical applications of UAV use in different habitat types for mammals as individuals, single species, or groups of many species at the same time [9]. In this review, we explore the traditional ecological monitoring techniques for wildlife and the condition of their habitats. We also discuss the evolution and growing applications of the UAV in wildlife conservation and management to tap into its potential for the development of an effective and efficient UAV-based ecological monitoring system. The review explores the achievements, challenges, and gaps in UAV application for wildlife conservation and management and discusses the efforts already undertaken by scientists to develop machine learning programs for automatic detection of mammals in the UAV-based images. Finally, we discuss the need for advanced research to generate inputs for the UAV-based ecological monitoring system. This data is key to the development of an integrated and effective ecological monitoring system in small community-based conservation areas, which are smaller than the core protected areas, such as national parks and game reserves. The synthesized information forms a basis for the potential uses of UAVs in small protected areas such as the WMAs in Tanzania. The conceptual flow chart below forms the framework of our review process (Figure 1).

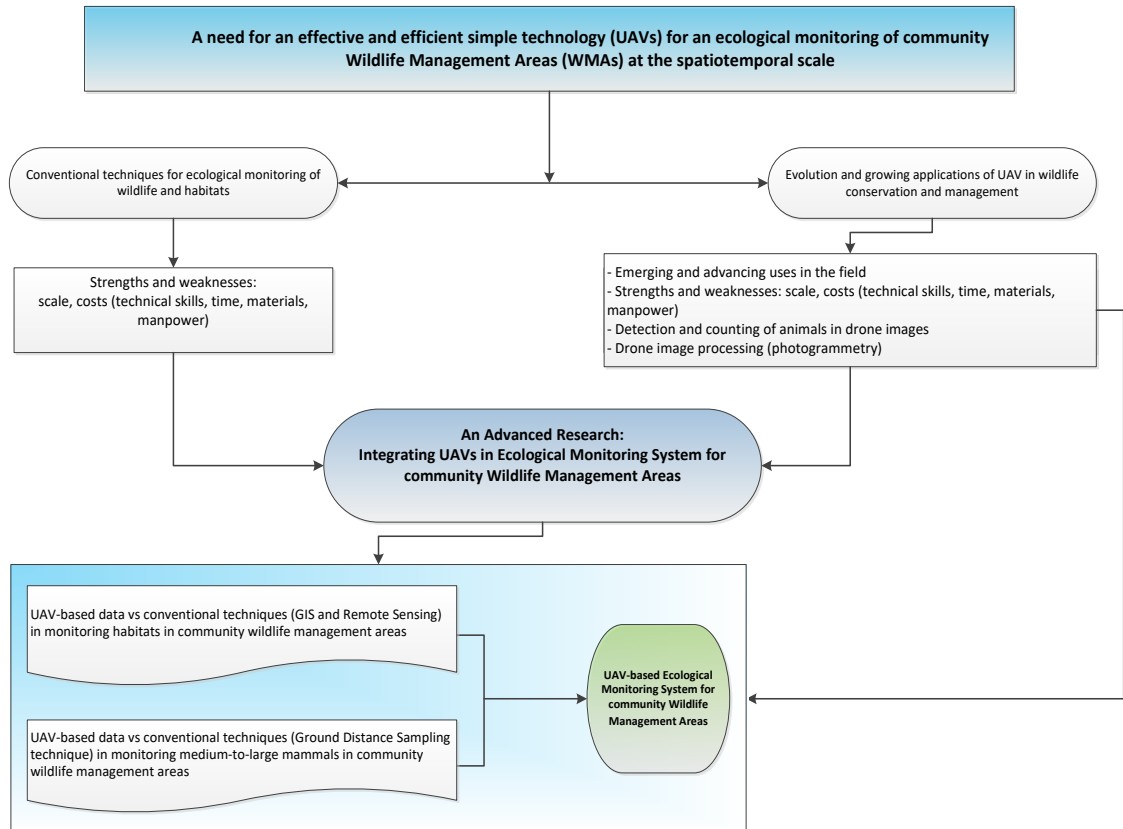

**Figure 1.** The conceptual framework for the review.

## 2. Conventional Techniques for Ecological Monitoring of Wildlife and Their Habitats

Many researchers and ecologists use conventional techniques in ecological monitoring of wildlife inside and outside protected areas. These include ground inventory techniques for flora and fauna species, geographical information systems (GIS) and remote sensing for mapping habitats and land use/land cover changes, and systematic reconnaissance flights (SRFs) for the aerial counting of

mammals [10–12]. However, these systems are faced with many challenges. For instance, they are costly, require much time and technical skills, and they are difficult to use in topographically-challenging areas [13,14]. Sighting probability is another factor that also limits the traditional aerial surveys for animals, as it tends to be less than 75% [4]. Furthermore, the aerial images obtained from traditional remote sensing tools such as MODIS, Landsat, and Quickbird have limitations in resolution compared to those obtained by the UAV-based remote sensing platform. Quickbird has the highest resolution, followed by Landsat and MODIS [10]. Furthermore, the cloud cover, orbit features of satellites, and restricted readiness of aircraft platforms impose limitations on temporal scale resolutions of most traditional sensors [13]. Organismic-level analysis constrains many satellites due to inadequate spatial resolution [10,15]. Those large satellite platforms providing high spatial resolution data up to 5 m are mainly operated on a commercial basis, making them costly [10]. With these technical gaps, the use of effective and affordable geospatial technology is recommended to obtain real-time images. The technology generates quality and reliable information for appropriate decision-making [16].

Despite the provision of real-time images at high resolutions by the UAV-based remote sensing aerial platform compared with the traditional remote sensing tools, ecologists and protected area managers need more information for the effective and efficient monitoring of medium-to-large mammals and their habitats at spatiotemporal scales. A clear understanding of the detection of different mammals and ecological attributes of their habitat conditions from the UAV images with different qualities captured at different flight heights and spatiotemporal scales is essential [2,16]. The nature and size of an area of interest may necessitate the use of UAVs in favor of traditional techniques [17]. For more than thirty years, modern photogrammetry and remote sensing have recognized the potential of UAV-sourced imageries for various uses in environmental and natural resources management [13,18]. The 21st century has witnessed the emergence and increasing application of new UAV technology in wildlife conservation and management [10,14]. International communities extended their uses from the militaries to civilians [19].

The UAVs with relevant capacity can address the technological challenges facing conventional techniques. Detection probability, high resolution, and operational skills are some of the challenges in the application of UAVs to collect relevant ecological monitoring data for wildlife management that its users of the UAV need to understand [3,14,20]. This technology can circumvent obstacles while flying and provides room for altitudinal adjustment [21] and is increasingly becoming a promising conservation tool in ecological monitoring [14,22–25]. The technology has the potential for quicker deployment than satellites or manned airplanes [10]. It is also useful for wildlife species that are sensitive to ground survey techniques [3]. The UAV-based data provides useful insights for timely management responses [26], and can greatly supplement the conventional techniques used in wildlife-based researches, ecological monitoring, conservation, and management practices [3,14,23,27,28]. Law enforcement units in Africa have also started applying UAVs in their operations in addition to animal detection and vegetation assessment [25,27,29]. Tanzania has inadequately applied the UAV technology in the wildlife sector, probably due to delays in the formulation of relevant legislations and regulations for its use. All of these call for detailed studies on its use in various heterogeneous wildlife habitats on a spatiotemporal scale before scaling it up to large protected areas [3,14,23,27,28].

## 3. UAV-Based Assessment of Animals and Their Habitats

Documentation on applications of UAV for both terrestrial and marine environments exists, as shown in Tables 1 and 2 [9,17]. A feasibility study testing the UAV technology in behavior research revealed its application in studying the crocodilian nests along the Kinabatangan River in Sabah, Malaysia [23]. The study managed to detect three nests at 200 m and 300 m flight altitudes without significant disturbance to the animals. However, the study focused along the river habitat and on one animal species only. The role of UAV in the monitoring of the restored small Oil river systems in the northwest of France has also been investigated [22]. The study focused along the riverine habitats strip and on single animal species only. In another study, a small UAV mounted with an infrared and

still cameras were used to test its ability in detecting and recognizing dugong and whales in Western Australia [7]. Other mammals studied in Australia using the UAV include the eastern grey kangaroo (*Macropus giganteus*) using video and infrared (IR) camera [2]; sharks, dolphins, and rays using ND4 circular polarizing filters; and Humpback whales using D90 12 megapixel digital SLR and standard definition electro-optical camera [7].

In North America and Europe, researchers have used UAVs mounted with different camera sensors to monitor mammals, birds, and reptiles. For instance, some of them used the near infra-red (NIR) sensor for sea turtles [28], while others used both RGB and thermal cameras for seagulls [30]. In Canada, other researchers used RGB and thermal infrared (TIR) sensors to detect and count deer using UAV. A combination of these sensors produced better detection and counting results. However, the study also focused on a single animal in less natural habitat. The detection characterization was conducted for the newborn deer (fawn deer) using UAV mounted by the thermal infrared camera in meadows of cultivated farmland [31]. In the study, individual animals were detected by the thermal infrared camera at 30 m flight height during cold weather. Despite the promising results, the study recommended further studies for improved fawn deer detectability using a UAV remote sensing platform.

In Sumatra, Indonesia, lightweight UAV showed different land uses and captured images of large mammals (Sumatran orangutan and Sumatran elephant) in areas neighboring Gunung Leuser National Park [14]. The study focused on the forested landscapes in the river ecosystem and areas with anthropogenic activities. Another study revealed the UAV's potential uses in environmental mapping and monitoring community-based forests in the tropics [32]. The study reported on many applications of UAV in forest plantations for mapping and monitoring forest fires based on scientific stands. Despite its potential usefulness in mapping and monitoring community-based forests, the study emphasizes strengthening the application of UAV in wildlife conservation and management.

**Table 1.** Examples of wildlife species monitored using UAV in Australia, Europe, and America.

| Class & Order | Species | Country | Monitored Aspects | UAV Class | Camera Sensor | Flight Height | Behavioral Response | References |
|---|---|---|---|---|---|---|---|---|
| Mammalia: Sirenia and Cetartiodactyla | Marine mammals (Dugong and whales, *Megaptera novaeangliae*) | Western Australia | Detection and recognition capability | Warrigul UAV | Colour/infrared/hi-res still photography | 1000 ft, 750 ft and 550 ft | Not recorded | [7] |
| Mammalia: Diprotodontia | Eastern grey kangaroo (*Macropus giganteus*) | Queensland, Australia | Response to UAV | DJI Phantom 3 Advanced | Video and IR camera | 120 m, 100 m, 60 and 30 m | Vigilance behavior | [2] |
| Chondrichthyes; and Cetacea | Sharks, rays and Dolphins | New South Wales, Australia | Detection and classification | DJI Phantom 4 | ND4 polarizing filters | 60 m | Not recorded | [5] |
| Mammalia: Cetartiodactyla | Humpback whales (*Megaptera novaeangliae*) | North Stradbroke, Australia | Detection | ScanEagle | (i) D90 12 SLR (ii) Standard Definition Electro-Optical Camera | 732 m | No significant reaction | [7] |
| Mammalia | White-tailed deer (*Odocoileus virginianus*), domestic dog (*Canis lupus familiaris*), domestic cat (*Felis silvestris catus*), | Canada, North America | Minimum detection heights | Quadcopter (SkyRanger) | Not provided | 1.5 m | Hearing response | [9] |
| Reptilia: Testudines | Sea turtles (olive ridley turtles); *Lepidochelys olivacea* | Costa Rica, North America | Population densities | Fixed-wing, senseFly eBee | Canon PowerShot S110 near-infrared (NIR) | 90 m | No behavioral change | [33] |
| Avian: Anseriformes | Flocks of Canada Snow Geese | Canada, North America | Population size | CropCama, a 2.5-m wingspan | Pentax Optio A20 infrared LED | 183 m (600 ft) | Not significant response | Chabot and Bird (2012) |
| Avian: Galliformes, and Anseriformes | Northern bobwhite upland game bird (*Colinus virginanus*), and mallard waterfowl bird (*Anas platyrhynchos*) | Canada, North America | Minimum safe flight heights | Fixed-wing platform (eBee; Sensefly) | Not provided | 1.5 m | No significant reaction | [9] |
| Mammalia: Artiodactyla | White-tailed deer (*Odocoileus virginianus*) | Germany, Europe | Detection and counts | ING Robotic Aviation | D7000 visible sensor (RGB) & Tau640 thermal infrared | 60 m | Not recorded | [4] |
| Mammalia: Artiodactyla | Roe deer (*capreolus capreolus*) | Germany, Europe | Detection and count | Micro air vehicle (MAV) Falcon-8 | Thermal camera core Tau640 | 30–50 m | Not recorded | [31] |
| Avian Charadriiformes | Seagulls | Germany, Europe | Counting seagulls colonies | Multicopters (Falcon 8, MD 4 1000) | Olympus PEN E2 (RGB and thermal cameras) | 15 m | They ignored the UAS | [34] |
| Avian: Charadriiformes | Black headed gull | Spain, Europe | Detection and recognition | Radio-control model aircraft, Poway, wingspan 1420 mm) | Panasonic Lumix FT-1, | - | Not provided | [35] |

Some countries in Africa have recently started using the UAV-based remote sensing aerial platform in different fields, such as wildlife conservation and management (Table 2). For instance, in Burkina Faso, the UAV was used to assess the population of large mammals in the Nazinga Game Ranch located in the south [36]. Large mammals, like the African elephant (*Loxodonta africana*), were easily detected and counted in UAV images when flown at the height of 100 m and 300 m without noticeable disturbances [36]. However, the study did not test for possible low flight heights at which the medium and large mammals could be detected and counted in the UAV images. In South Africa, the UAVs were used to predict densities of different wildlife species and modeling cattle in Kwa Zulu-Natal Province [37]. It attempted to fill the gap of UAV's applications in assessing the impacts of infrastructure on wildlife species, particularly in the collision hazards of the drone for birds. Interestingly, the study used cattle as a proxy for the UAV-based prediction of the spatial distribution of ungulates in protected areas. We noted that variations and challenges exist when using real wild ungulates in their heterogeneous habitats. In Rwanda, the potential of UAVs in creating and updating maps has also been reported [13].

In Loango National Park, Gabon, studies showed the positive applicability of the UAV technology in locating chimpanzee nests and identifying fruiting trees [38]. The study tested four factors as predictors of nests detection: heights of nests locations on trees, canopy openness (either open or closed), and type of habitats where the nests were detected as well as their ages. However, the study did not take into consideration the habitats' heterogeneity and temporal scales. The spatial heterogeneity of habitats has an influence on the population status and distribution of grazing and browsing mammals in landscapes at temporal scales [8,38]. Complex human-environmental interactions at different spatiotemporal scales also pose major challenges, such as in human-dominated landscapes [39,40]. A study in the Democratic Republic of Congo used an RGB sensor to estimate the population status of the common hippopotamus (*Hippopotamus amphibious* L.) [30]. The population status (number and structure) of the Nile crocodile (*Crocodylus niloticus*) was also studied using UAV in South Africa [37].

In western Tanzania, a few studies have used the UAV technological aerial platform in the wildlife sector. Among these is the study that was carried out by Jane Goodall Institute (JGI), which used the UAV to survey chimpanzee nests and focused on the feasibility of UAV to participatory community forest monitoring strategies [41]. The study assessed the possible factors that were likely to influence the nests' detectability. The vegetation, seasonality, image resolution, nest height, and color of the nests were tested as possible influencing factors. However, this study also focused on one animal species and two habitats (forest and woodland) only. Another study conducted in Tanzania revealed that the detection probability of both Thermal IR and RGB sensors was significantly affected by the canopy density in a landscape [8]. Another study that used UAV to assess wetland habitats and human activities in Tanzania recommended the integration of UAV aerial platforms in wildlife conservation and management [1]. Generally, the majority of the UAV-based studies have focused on a single or a few individuals of animal species and homogenous habitats with less diverse vegetation attributes at a small spatiotemporal scale.

**Table 2.** Examples of animal species monitored using UAV in Asia and Africa.

| Class and Order | Species | Country | Monitored Aspects | UAV Class Used | Camera Sensor | Flight Height | Behavioral Response | References |
|---|---|---|---|---|---|---|---|---|
| Mammalia: Primates, and Proboscidea | Sumatran orangutan and Sumatran elephant | Sumatra, Indonesia, Asia | Detection | Hobbyking Bixler | Canon IXUS 220 HS (Pentax) and GoPro Optio WG-1 GPS | 100 m, 180 m | Not recorded | [14] |
| Mammalia: Carnivora | Endangered species: Tiger | Nepal, Asia | Endangered species against poachers | GPS-enabled FPV Raptors | Still and video cameras | <200 m | Not recorded | [42] |
| Mammalia | Large mammals | Namibia, Africa | Detection and identification | Canon PowerShot S110 compact camera, | Red Green and Blue (RGB) bands | - | Not recorded | [43] |
| Mammalia: Proboscidea | African Elephant (*Loxodonta africana*) | Bukina Faso, Africa | Animal counts | Gatewing6100TM | Ricoh GR III camera | 100 m and 300 m | No reaction | [36] |
| Mammalia: Perissodactyla | Black rhinoceros (*Diceros bicornis*) and the white rhinoceros (*Ceratotherium simum*) | South Africa | Rhinoceros and poachers detection | Fixed-wing, Easy Fly St-330 (St-models, China) | (i) FPV video camera, (ii) Panasonic Lumix LX-3 digital camera (iii) thermal video (IR camera module) | 32–180 m | No flight response | [27,43] |
| Mammalia: Artiodactyla | Common Hippopotamus (*Hippopotamus amphibius* L) | Democratic republic of Congo, Africa | Population | Fixed-wing, Falcon UAS | Sony Nex7 digital still camera (RGB) | 20 m, 40–150 m | No reaction from 40 m | [28] |

**Table 2.** *Cont.*

| Class and Order | Species | Country | Monitored Aspects | UAV Class Used | Camera Sensor | Flight Height | Behavioral Response | References |
|---|---|---|---|---|---|---|---|---|
| Reptlia: Crocodilia | Nile crocodile (*Crocodylus niloticus*) | South Africa | Population | DJI Phantom 3 Standard Drone | Still camera of 1/2.3″ sensor, fast f/2.8 prime lens | 40 m, 55 m and 70 m | Not recorded | [37] |
| Mammalia: Primate | Chimpanzee (*Pan troglodytes*) | Tanzania, Africa | Nests detection and population estimate | (i) Skywalker X5 frame; (ii) HBS Skywalker 100 km Long Range Fix Wings drone | (i) Canon S100 camera operating a CHDK firmware modification; (ii) Sony RX100M2 | 90 m | Not recorded | [41] |
| Mammalia: Proboscidea | African Elephant (*Loxodonta africana*) | Tanzania, Africa | Human-Elephant Conflict control | DJI Phantom 2 | No camera | 50 m | Flight response | [44] |
| Mammalia: Primate: Primate | Human poachers | Tanzania, Africa | Detection of poachers | Multicopter (DJI F550 frame) | (i) 16 MP RGB Survey 2 camera; (ii) Thermal Capture v1.0 TIR camera | 70 m, and 100 m | Not recorded | [8] |
| * Wetland | Wetland habitats | Tanzania, Africa | Wetland condition and human threats | Multicopter DJI Phantom 4 Pro | 20MP camera with 1 in sensor CMOS with a Field of View of 84.4° and a lens of 8.8 mm equivalent to a 24 mm | Not provided | Not applicable | [1] |

* This was purposively added as an example of special wetland habitats assessed using UAV.

The detection and identification of animals in the UAV-based images is challenging, depending on the type and size of the sensor used and flight heights [36]. This review found that it is important to understand the factors to be considered when flying the UAV. In order to fulfill this need, a number of publications on the efforts made by scientists to come with machine learning programs for automated detection and counting animals in the drone images were reviewed. For instance, some authors tested state-of-the-art object detection using the shape and color of the objects [45]. The workers reported color descriptors (robust hue descriptor, opponent derivative descriptor, and color names) as the main useful attribute for the detection. However, the experiment was not extended to the natural environment to test applicability to wildlife and their habitats. Another state-of-the-art generic object recognition method used distinct animal detection in single images and experimented with the deformable part-based model (DPM) and exemplar support vector machine (SVM) together. These techniques were found to be promising, though challenging, for the detection and counting of individuals in large herds of animals [46–50].

We also found that the training of a large, deep convolutional neuron network for animal detection in the UAV images performs better than the state of the art [51–54]. However, it faces similar challenges in dealing with mammals in medium to large herds. A course-to-fine approach is useful for a few objects to speed up the detection of deformable objects in drone images [55]. The literature suggests that the uses of UAV and computer vision structure-from-motion (SFM) algorithms are effective in measuring forest canopies [56]. The experiment focused on estimating forest structure only, leaving aside the wild animals and their habitat quality attributes at different spatiotemporal scales. The experimented multi-criteria object-based image analysis (MOBIA) for multispecies detection was found effective for detecting only one species [57]. It did not detect and classify individuals of several species concurrently. However, it detected large mammals such as African elephants. The detection and counting of large mammals using the manual technique in the drone images are still useful depending on the sensor capacity and flight height. However, the technique is challenging because of the time and the large manpower involved [36].

Regarding the use of a UAV-based remote sensing platform in surveying different vegetation types, many of its recorded applications have focused more on crops and plantation fields than in rangelands and natural forests [58]. The studies have not put much emphasis on the use of NDVI values in determining photosynthetic levels of wildlife forages in habitats at spatiotemporal scales.

## 4. A Need for an Integrated UAV-Based Ecological Monitoring System

The need for appropriate ecological monitoring of conservation areas dates back to when humans began replacing their areas with protected area establishments [31]. The Convention on Biological Diversity (CBD) emphasizes on the monitoring conservation initiatives, such as the community wildlife management areas [28]. Monitoring is also needed by conservation communities, donors, policy, and decision-makers to improve conservation actions and management practices. The development of an effective and efficient integrated UAV-based ecological monitoring system that generates real-time ecological data is crucial in complementing the conventional survey techniques [59].

Sustainable wildlife management and conservation is a function of reliable ecological data for monitoring animals and their habitats [60]. The UAV technology has the potential of fulfilling this ecological demand [21,45,61]. It enhances the advancement of aerial imagery in science since it generates higher quality and more accurate data than conventional techniques [45,62]. Many studies support the need for ecological monitoring systems that use technologically-improved methods for efficient and effective monitoring of wildlife and their habitats [28,31,63,64]. This need forms the basis for advanced research for generating evidence-based data as inputs for developing an integrated UAV-based ecological monitoring system for wildlife and their habitats in community wildlife management areas in Tanzania.

Real-time ecological data about the abundance and distribution of mammals and their habitat conditions are core values of the UAV-based wildlife monitoring system [4,14,16,26,29].

Habitat conditions indicators include the level of degradation, encroachments (human settlements, farms, and cattle shelters), vegetation cover, wildfires, and logging. It is argued that an appropriate collection of ecological data based on a well-developed system in line with a good conceptual model of an ecosystem or population is important [65]. Since the UAV is a new aerial platform in wildlife conservation and management, it is important to wildlife managers, researchers, decision-makers, and other related practitioners. As stakeholders, they have to understand what and how much the UAV technology has successfully contributed to conservation and how else it can be made much more useful in ecological monitoring [3]. If well used, it will improve our capability to monitor ecological dynamics in landscapes [26,66]. The technology saves time and can be deployed many times in challenging places, enhancing safe data collection [67,68]. Appropriate understanding and monitoring of the impacts of various threats to ecosystems and their biophysical resources requires the integration of emerging technologies, such as the UAVs over the traditional field survey techniques.

There is scanty information on UAV-based ecological monitoring of medium-to-large mammals, particularly for groups of more than one species and many individuals of the same species in heterogeneous habitats. Many studies have focused on a single or few species, and few individuals in homogenous and less diversified habitats. Much concern is on heterogeneous ecosystems where traditional techniques such as field plots may be used, thereby generating inappropriate ecological data [59]. A majority of protected areas in tropical landscapes are characterized by complex habitats that call for advanced technologies, such as the UAV-based remote sensing tool aerial platform, that can generate diverse data for appropriate ecological monitoring of animals in such heterogeneous habitats (source). The technology, if well explored for ecological monitoring of WMAs that are utilized by both migratory and non-migratory mammals in different seasons, will be a much more reliable tool for local communities, managers, and researchers. For instance, in the Tarangire-Manyara ecosystem in northern Tanzania, a majority of mammals tend to concentrate inside the park, along the Tarangire River as the primary source of water for migratory ungulates during the dry season [69]. The ecological survey of mammals, if conducted during the dry season only, may misinform wildlife utilization planning, policy, and decision-making [26]. Furthermore, the challenges associated with the conventional techniques signals a need for a cost-effective UAV-based aerial platform. This is important for the community WMAs if it can be afforded and used by the managers to capture high-resolution images from which real-time ecological data for the animals and habitats are obtained. The local communities and managers of the WMAs need to have appropriate UAVs-based ecological monitoring systems that suit their needs.

## 5. Relevance of UAV Technology in Ecological Monitoring of Small Community Wildlife Management Areas

Sustainable conservation and management of wildlife inside and outside core protected areas, such as the WMAs in Tanzania, is largely anchored on reliable ecological monitoring data. In Kenya, inadequate ecological management of the conservation areas has been linked with deficiencies in ecological information, ultimately leading to loss of wildlife [40]. The researchers call for serious ecological monitoring of multi-species and their habitat variables. This has an implication for the application of a technologically effective and efficient ecological monitoring system. The UAV technology, as the newly emerged remote sensing aerial platform, has significant potentials for sustainable conservation of wildlife in the community wildlife management areas.

The WMAs are a new protected area category in Tanzania formed on village land whereby neighboring villages contribute part of their land to form a jointly managed WMA [70,71]. The establishment strives to ensure conservation and management of wildlife, biodiversity, and habitats on community lands. The WMA's concept is a strategy for wildlife conservation outside core protected areas while supporting the National Strategy for Growth and Reduction of Poverty of 2005 [72]. This is also in line with the Communal Areas Management Programme for Indigenous Resources (CAMPFIRE) program in Zimbabwe designed to tackle environmental problems and help rural

communities to manage their resources for their development [73–77]. The approach was based on the rationale that many wildlife and biodiversity found outside core protected areas face many threats, including illegal harvest and habitats loss, particularly buffer zones, dispersal areas, and migratory corridors. The WMAs, thus, serve as strategic buffers, dispersal areas, and/or migratory corridors for the wildlife [72,78]. These areas are jointly managed through the establishment of a Community Based Organization that will later be upgraded to Authorized Association (AA) status by the Ministry of Natural Resources and Tourism once they have accomplished all the requirements stipulated by the Wildlife Management Area Regulations of 2012 [71].

Despite all the aforementioned conservation policy efforts to establish community conservation areas, problems and challenges still persist on their ecological monitoring for appropriate actions to address them. Ecological monitoring in the WMAs is especially challenging due to the lack of a quick, effective, and technologically user-friendly monitoring system that can be applied by both researchers and locally-based monitoring teams to generate useful real-time data as an alternative to the traditional biodiversity and habitats assessment techniques [79]. These ecological monitoring gaps form the basis for this review to ascertain how much ecologists, researchers, and local communities can rely on the UAV technology for ecological monitoring of wildlife and habitats in the community conservation areas. Inadequately monitored wildlife corridors and dispersal areas, including those embedded in the community WMAs, result in the decline of wildlife populations [80].

Effective monitoring of animals and their habitats detect changes that are taking place and the quality of biodiversity, informing decision-makers to address the possible causes of the changes [81]. Monitoring is the information system that deals with the observation, estimation, and forecast of changes in the environment, created to allocate the anthropogenic component of these changes on the background of natural processes [82]. It is the "process of repeated observations (for specific purposes) of one or more elements of the environment according to pre-arranged schedules in space and time" [83,84]. Monitoring data can indicate any need for correction of the management strategies towards achieving the intended objectives [84]. Ultimately, such data is useful in improving conservation and management practices and the performance evaluations of the community WMAs.

## 6. Challenges Associated with the Use of UAV in Conservation

### 6.1. Legal and Ethical Requirements

The recent emergence of UAV technology and its civilian uses have sparked legal and ethical regulatory authorities in many countries [9,17,21,85–88]. The permit requirements, including visual sine of sight (VLOS), reduce effective uses of the UAV in environmental and conservation research works [60]. The UAV has long been used in military missions until recently, when the civilians were allowed to purchase and use them for different purposes, such as conservation, medical, commercial, and social activities [89]. There are legal and ethical considerations regulating its use following the International Civil Aviation Organization (ICAO) established by the International Civil Aviation Convention (Chicago Convention) in 1994 for safety, privacy, and data security [28]. Five key aspects are considered by many countries when setting regulations for appropriate use of the drones [21]; (i) the type, size, and intended use, (ii) the geographical locations for its use in relation to restricted areas such as military places, (iii) adherence to specific legal or administrative procedure pertaining to specific country, (iv) necessary drone's technical information in relation to its mechanical, control, and communication aspects, and (v) adherence to ethical issues at the operation areas. In addition, operational, age, safety, and legal aspects are reflected in the pilot's license [13]. The potential threats of the UAV on human privacy due to cameras and infrared sensors that enable identification and recognition of objects, including humans, if captured, were reported [13].

Accordingly, the process for acquiring government permits for flying UAVs in most countries takes much time due to substantial restrictions imposed on its official uses [9,28,85,90]. For instance, in Tanzania, a permit is provided by the Tanzania Civil Aviation Authority (TCAA), which has a

minimum set of rules and regulations to comply with before operating the UAV (www.tcaa.go.tz). Furthermore, the permit directs the applicants to seek another permit from the military that provides a Military officer for fieldwork at the researcher's costs. All of these steps have significant time and cost implications. Hence, any design of UAV-based research has to consider timely applications for government approvals. Generally, [90] advises that

> "If ecologists hope to realize the potential for advances in aerial imagery, population and community ecology, and large-scale conservation that can result from using UAV technologies, we have to advocate for lower barriers to entry so UAVs may become part of the ecologist's "toolbox". The status quo of governmental regulation of UAV-driven research requires effort and time beyond what is realistic for practitioners who wish to use the UAVs as an additional element of a research program".

*6.2. Technical Challenges*

It is reported that the application of small UAVs does not need sophisticated skills [3,14,20], and relatively little training is needed [24,26]. However, the necessary and basic technical skills are inevitable to ensure safe operations and image processing to generate information in fulfilling pre-determined ecological monitoring objectives [21]. The author(s) emphasize(s) the importance of addressing any technical challenges well in advance before flying the drone. For example, appropriate flight mission planning and executions are crucial [11,88]. Other technical challenges worth knowing that are associated with the use of drones include power limitations and low flight time aspects that can affect its performance [21]. Yet, many developing countries, including Tanzania, have inadequate capacities for application of UAV in professional ways for obtaining quality images and undertaking photogrammetric processing for conservation. For instance, it is worth knowing that accurate positions and flight heights or altitudes, for any UAV used, influences the sensor's accuracy of the measurements obtained [87,88].

Drone imagery photogrammetric processing is also challenging due to variations in different factors: levels of image overlap and relief displacement concerning flight heights attributed to topographic relief differences in landscapes [11,13,91,92]. Furthermore, it reported that small, a low-cost UAVs tend to have accuracy limitations of the Global Navigation Satellite System, which requires Ground Control Points for ensuring image processing accuracy [13]. The author(s) provide(s) further precautions that insufficient overlap during image acquisition can cause digital structural model (DSM) deformation, requiring appropriate flight mission planning, image acquisition, software, and algorithms to be used.

*6.3. Weather Conditions*

Different weather conditions have different levels of impact severity on drone flight performance. Cloud cover, fog, haze, and glaze cause moderate impact, while wind and turbulence, rain, temperature, humidity, snow, and solar storms cause adverse impacts. Severe impacts on drone performance are mainly caused by hurricanes, lightings, halls, and tornadoes [93]. Solar noon hours are worth considering for animals and vegetation photographs to avoid shadow effects during orthomosaics processing [94]. The solar noon hours differ from place to place and day-to-day. Hence, one should be able to visit sunrise websites for solar hours before flying and taking drone images.

## 7. Conclusions

This review reveals that the use of a UAV-based remote sensing platform in wildlife conservation and management is increasing due to its usefulness in generating real-time data for timely decision making. Many traditional techniques for ecological surveys of wildlife and their habitats have many challenges like the high costs, considerable time, large number of personnel, and high level technical skills involved, as well as inaccessibility to topographically-challenging habitats. The use of UAV technology is becoming increasingly popular for various civilian applications. If well experimented

at the spatiotemporal scale for generating diverse ecological data for more than one species of medium-to-large mammals found in groups or individually in mixed habitats, it can become a much more reliable tool. It will be a promising tool in addressing the challenges associated with the traditional techniques in challenging yet heterogeneous wildlife habitats. Monitoring of animals in mixed habitat types, such as open grassland, wooded grassland, shrubland, open and closed woodlands, lakeshores, and forests, needs an effective and integrated system that is capable of capturing diverse and quality data on the animals and habitats condition attributes. The mixed habitats have implications on animal visibility and vegetation reflectance depending on the sensors used and flight heights selected. Hence, there is a need for a UAV-based ecological study in heterogeneous habitats at flight heights that can enable the detection and counting of medium-to-large mammals and provide habitats' condition attributes with insignificant disturbances to animals.

**Author Contributions:** Conceptualization, L.J.M.; methodology, L.J.M.; investigation, L.J.M.; writing—original draft preparation, L.J.M.; writing—review and editing, L.K.M., and P.A.N.; supervision, L.K.M. and P.A.N.

**Funding:** This research was funded by the Centre for Research, Agricultural Advancement, Teaching Excellence and Sustainability in Food and Nutrition Security (CREATES–FNS).

**Acknowledgments:** We are grateful to The Nelson Mandela African Institution of Science and Technology (NM-AIST) for appropriate academic arrangements that guided, administered, and provided mentorships without which this review manuscript could not have been accomplished.

**Conflicts of Interest:** The authors declare no conflict of interest. The funders (CREATE) had no role in the design of the study; in the collection, analyses, or interpretation of data; in the writing of the manuscript, or in the decision to publish the results.

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
