# Peer review of "Integrating UAV Technology in an Ecological Monitoring System for Community Wildlife Management Areas in Tanzania"

_sustainability, doi:10.3390/su11216116_

Round 1

Reviewer 1 Report

Very timely and comprehensive review which will be a very useful contribution to the field

Reviewer 2 Report

Revision of the manuscript sustainability-593170 by Mangewa et al. "Integrating UAVs Technology in Ecological Monitoring System for Community Wildlife Management Areas in Tanzania"

The manuscript by Mangewa et al. "Integrating UAVs technology in ecological monitoring system for Community Wildlife Management Areas in Tanzania" is an extensive and in-depth review of the use and potential of UAV technology in the field of wildlife monitoring aimed primarily at the management of problem areas. The Authors have analyzed in detail a large number of studies published in international journals and carried out in different countries distributed in almost all continents. Also the spectrum of species considered is high and this give value to the work. I personally think that the review is very useful in the field of wildlife management and not only, and that it demonstrates how this technology can be adopted in a more extensive and implemented way.

The only flaw I found in the manuscript is the total lack of tables and graphs that summarize the results of the bibliographic research. For example it would be interesting to show the trend over the years of studies using UAV technology, their distribution by continents and countries, the frequency of use on different species, how many species are monitored simultaneously, the frequency of studies that at the same time use UAV technology for the simultaneous monitoring of habitat changes, how many studies have been carried out at local scale and how many at larger scales, etc. All these aspects are covered in the review but some graphs and / or tables could make the readability of the text easier and the understanding more immediate. It is also easier to remember a graphic than a text.

Reviewer 3 Report

I found the bases for this article interesting, and agree with the authors that there is a significant need for further research and development in this field. However, I struggled to retain interest as I read deeper into the manuscript because the structure and grammar are very poor, and I suggest that substantial revisions are required to make a case for publication on the basis of the English alone. Throughout the manuscript the English is clumsy and often inhibits understanding of the concepts presented by the reader. Several sentences needed to be re-read multiple times before I was able to understand their meaning, and for some sections I still don’t understand exactly what the authors were trying to communicate.

Further to my language concerns, I have issue with this article being labelled as a review. I don’t feel that it really interrogates the international literature sufficiently to deserve the title, and is not exhaustive enough in its backgrounding and contextualisation. I was able to identify easily a dozen or more useful and potentially applicable papers on fauna monitoring that I am familiar with from previous work that are not cited. While some of these are focused on birds or marine animals and do not align specifically with the articulated aims of this study, they may still be useful to mention as they show the wide range of fields in which UAV monitoring can be applied. Additionally, some of the papers not cited are considered among the more important papers on fauna monitoring with UAVs, as they use more advanced sensors such as thermal cameras (e.g., ‘A UAV-based roe deer fawn detection system’ and ‘Visible and thermal infrared remote sensing for the detection of white-tailed deer using an unmanned aerial system’), or prove that a lack of behavioural response does not necessarily indicate a lack of physiological response (e.g., ‘Bears Show a Physiological but Limited Behavioral Response to Unmanned Aerial Vehicles’). There are also several previously and recently published reviews on UAV use in adjacent fields that may also be useful to cite. In particular, I found the lack of discussion of different sensors disappointing given that they are mentioned in a flora section, which isn’t even the main thrust of the article. I almost found the current structure of this article more closely aligned to that of an opinion article rather than a review, except that it lacks a strong core argumentative theme. The authors should go more in depth with the papers they cover, and provide greater discussion of the ethical concerns of using UAVs to monitor fauna may also be warranted.

Finally, greater care needs to be taken with the referencing. For example, references 3, 6, and 48 are the same paper, as are 2 and 32.

Round 2

Reviewer 3 Report

The authors are thanked for their comprehensive revision of the manuscript, which is now much improved and presents a significant body of information for the literature. I still have some specific minor concerns about the presentation and language/grammar of individual sections, but do not have the time to identify and address these in detail - I suggest the editor liaise with the authors to ensure the quality of the English is sufficient prior to publication and for this reason I have suggested accept pending minor revisions. I acknowledge the point by the authors that an external English speaker has revised the MS and certainly the quality throughout is much improved - however, it could still be improved markedly in some sections prior to publication.